# Development and preliminary validation of infrared spectroscopic device for transdermal assessment of elevated cardiac troponin

Jitto Titus[1], Alan H. B. Wu[2], Siddharth Biswal[1], Atandra Burman[1], Shantanu P. Sengupta[3] & Partho P. Sengupta[4✉]

## Abstract

**Background** The levels of circulating troponin are principally required in addition to electrocardiograms for the effective diagnosis of acute coronary syndrome. Current standard-of-care troponin assays provide a snapshot or momentary view of the levels due to the requirement of a blood draw. This modality further restricts the number of measurements given the clinical context of the patient. In this communication, we present the development and early validation of non-invasive transdermal monitoring of cardiac troponin-I to detect its elevated state.

**Methods** Our device relies on infrared spectroscopic detection of troponin-I through the dermis and is tested in stepwise laboratory, benchtop, and clinical studies. Patients were recruited with suspected acute coronary syndrome.

**Results** We demonstrate a significant correlation ($r = 0.7774$, $P < 0.001$, $n = 52$ biologically independent samples) between optically-derived data and blood-based immunoassay measurements with and an area under receiver operator characteristics of 0.895, sensitivity of 96.3%, and specificity of 60% for predicting a clinically meaningful threshold for defining elevated Troponin I.

**Conclusion** This preliminary work introduces the potential of a bloodless transdermal measurement of troponin-I based on molecular spectroscopy. Further, potential pitfalls associated with infrared spectroscopic mode of inquiry are outlined including requisite steps needed for improving the precision and overall diagnostic value of the device in future studies.

### Plain language summary

The number one cause of death in the US is heart disease. With 10 million patients visiting the emergency departments in a year with chest pain, 8 million are unrelated to cardiac issues. This places a burden on hospitals leading to suboptimal patient outcomes. In patients with cardiac issues, the time clinicians take to intervene dictates reversible or irreversible heart damage. However, current markers used to test for cardiac issues require blood sampling, limiting access to and frequency of testing. This study introduces a non-invasive cardiac marker measurement device without any form of blood draw, based on measurements taken by a wearable device through the skin. Preliminary studies show high conformance to the standard of care technologies, indicating that the technology has potential to enable more rapid, frequent, accessible and non-invasive detection of cardiac issues such as heart attacks.

[1] RCE Technologies, 233 Arnold Mill Rd Suite 300, Woodstock, GA 30188, USA. [2] UCSF, 1001 Potrero Ave, San Francisco, CA 94110, USA. [3] Sengupta Hospital and Research Institute, Amravati Rd, Ravi Nagar, Nagpur, Maharashtra 440033, India. [4] Division of Cardiology, Rutgers Robert Wood Johnson Medical School, 1 Robert Wood Johnson Place, New Brunswick, NJ 08901-1311, USA. ✉email: partho.sengupta@rutgers.edu

Over 10 million patients present with chest pain[1] in emergency departments (ED) in the United States alone. Over 80% of these are due to non-cardiac causes, resulting in an unnecessary burden in the ED, revealing the need for an instant non-invasive screening technique that can streamline the ED workflows[1]. Furthermore, 1 out of 5 myocardial infarctions (MI) is asymptomatic (silent), leading to nearly 200,000 silent MIs each year in the US[2]. Therefore, the development of new technologies that can allow early non-invasive detection of myocardial injury is imperative.

Detection of cardiac troponins[3] to assess cardiac injury has been around since the 1990s. The state-of-the-art[4] troponin quantitation mechanism is based on immunoassays[5] involving the use of two or more antibodies, one of which is labeled, typically with a chemiluminescent tag, which adds another level of complexity[6] in the analysis. While immunoassays are highly developed and accurate, extensive sample preparation including cumbersome blood[7] draws are required. Furthermore, it mandates the logistics between physician and laboratory. Point-of-care (POC) assays are becoming increasingly available such as Abbott iSTAT[8] but suffer from low sensitivity making them incompatible with rapid rule-out algorithms[9]. High sensitivity immunoassays have advanced state of art possibilities for point of care and home healthcare modalities with innovations by startups like Luminostics[10]. Similarly, recent developments with microneedle patches[11] further demonstrate promise for longitudinal monitoring of the levels of inflammatory biomarkers. While the recent POC[12] solutions reduce time to test results, there still remains a dependency on blood draw coupled with lower analytical sensitivity compared to central laboratory testing, hence limiting their application toward effective discharge from the ED. Posited in this letter as a solution is infrared spectroscopy[13], a widely used and applied characterization technique due to its ability to probe into the material at the molecular level and hence an inherently sensitive mode of interrogation. The most appealing aspects include the benefit of minimal or no sample preparation required. There are innovative wearable devices based on functional-near-infrared light interrogation[14] such as, cortical hemodynamics[15], blood oxygenation[16], blood glucose monitoring[17], etc. However, these methodologies are not sensitive to the molecular composition of materials and therefore require labeling or tagging the molecule of interest for its detection. Molecular spectroscopy is one of the cornerstones of analytical tools used to study structural and compositional chemistry[18]. This technology typically involves the study of the interaction between mid-infrared (MIR) radiation with matter for which, as a diagnostic tool in a wearable format, there is no device yet to-date that has shown potential. There have been some endeavors leveraging the diagnostic capabilities of MIR with analytes such as in serum[19], urine[20], breath[21], skin[22] etc. Being well established as an indispensable tool in material science[23], infrared spectroscopy has been since applied in food, drug, environmental, forensics disciplines and importantly in the biomedical field such as cancer detection[24], and even cardiac care[25,26]. Another advantage[27] is that, testing using non-invasive devices like pulse oximeter, breathalyzers, and bilirubinometers, are exempt from CLIA regulations because a sample is not taken from the body. Recently, complex MIR spectrometers have been scaled down in footprint to fit into portable-sized benchtop devices such as the MZ5 by OceanInsight. However, there is no non-invasive wearable diagnostic device based on MIR spectroscopy. IR spectroscopy has been elusive[28,29] as an alternative to current POC solutions due to the following reasons[30]: (a) Signal to noise ratio strongly dictates the minimum detectable limit (b) Since all matters are sources of infrared radiation, efficiencies of IR-based devices can be confounded by stray light (c) The most sensitive mode of operation which is Fourier Transform IR spectroscopy requires a large footprint and is highly sensitive to mechanical vibrations, due to moving components thus often confined to ex vivo modalities. This correspondence outlines steps taken to circumvent or mitigate the mentioned challenges leading to an efficacious non-invasive device capable of risk-stratifying[31] ACS[32] patients based on Troponin-I (cTnI) levels.

## Methods

The bloodless transdermal infrared spectroscopic device to assess elevated troponin-I was developed over four research phases.

**Ex vivo exploratory research**. First, an investigational study was conducted to determine the spectral features that are unique to cardiac markers such as cardiac Troponin I (Sigma-Aldrich, T9924-20UG), creatine kinase-MB (Sigma-Aldrich, C0984-100UG, and B-type natriuretic peptide (Sigma-Aldrich, B5900-.5MG), with the optical characterization of these substances in their pure form. A Nicolet ThermoFisher IS50 infrared spectrometer (IEN Labs, GaTech) employing a (single-bounce) diamond IRE, is used to identify a spectral signature for cardiac troponin. This allows one to optically detect and quantify the presence of that biomarker in a host substrate such as whole blood. De-identified healthy whole blood was procured and characterized to determine if there are any confounding overlaps in the absorbance peaks of blood and cardiac biomarkers.

Consequently, to confirm the efficacy of the ATR mode of interrogation, 30 biologically independent de-identified blood samples with the corresponding measurement of high sensitivity-cTnI values were procured in collaboration with the department of laboratory medicine at UCSF upon executing a Human Material Transfer Agreement. These blood samples that had already been collected prior to this study in accordance with applicable laws, regulations, patient consent forms and authorizations pursuant to Institutional Review Board. Further ethics approval and consent were not required as the samples were de-identified and anonymized in accordance with the Health Insurance Portability and Accountability Act. Blood samples included were those identified with a spectrum of Troponin-I values between the limit of detection (LOD) and upper reporting limit of the SOC assay namely, Advia Centaur[33] Siemens hs-cTnI. The blood samples were optically characterized ex vivo with the modality of total internal reflection using Nicolet ThermoFisher IS50 Fourier Transform infrared spectrometer (IEN Labs, GaTech) employing a (single-bounce) diamond IRE. Each blood sample to be characterized is one microliter in volume as deposited on the IRE. Each sample was measured in triplicates, with every repeat being an average of 32 co-added scans at a resolution of $4\,cm^{-1}$.

**Development of a transdermal kit**. A benchtop Attenuated Total Reflectance (ATR) based spectrometer by OceanInsight was implemented in a Cardiac care setting in collaboration with West Virginia University, Heart & Vascular Institute. It was hypothesized that transdermal measurement in the indirect correlation with myocardial injury would provide insights toward possible troponin measurement capability. An optical measurement was performed on the thumb of 4 normal and 5 cardiac patients. A multi-variate cluster analysis was performed on this data employing Ward's algorithm with squared Euclidean method focused on the wavelength range including some of the unique absorbance features relating to Cardiac Troponin-I with an interrogation window of 1.7 μm. The specificity of this window to cTnI as opposed to cTnT and other troponin isoforms is to be

investigated in future studies. This unbiased heterogeneity analysis classifies data based on similarity.

**Feasibility of point-of-care assessment**. ATR-FTIR although proven efficacious, is challenging as a candidate for POC devices due to its large size and susceptibility to mechanical vibrations. Typically with FTIR, a wavelength sweep is done by moving the mirrors of an interferometer and consequently transforming the data from a spatial to frequency domain. However, there isn't a need for a wavelength sweep as the wavelength ranges of interest, based on the ex vivo characterization of cardiac biomarkers, are previously determined. This allows for the selective sensitization of the infrared detector to the wavelengths corresponding to cardiac troponin-I by means of interferometric optical filters. As the next developmental step, an ambulatory non-invasive transdermal wearable device was deployed toward a pilot study conducted in the cardiac observation unit by recruiting patients with suspicion of ACS upon obtaining proper consent to obtain 24 biologically independent samples. The wrist wearable was installed on the patient's wrist within 10 min of a SOC blood draw, with the cTnI levels reported using the CE certified Snibe Maglumi-1000 high sensitivity assay. This version of the optical sensor was designed to be portable (palm-sized) by using a broadband infrared light source, germanium IRE and, thermopile detector with specifically chosen filters sensitive to two optical ranges: the first is representative of the Amide II band which is used as an internal normalization reference, the second range such that cTnI would have the largest contribution to the absorbance. Due to the absence of moving parts and complex optical components, the small form factor was achieved while minimally affected by mechanical vibrations. The confounding effect of stray light was negated by pulsing the emitter at 4 Hz while polling the thermopile at 8 Hz thus recording both the on and off state of the emitter. A differential of these two states accounted for the extraneous light captured by the detector.

The pipeline used for the data analysis schema toward establishing a correlation between the proposed noninvasive device and standard-of-care data is illustrated in Fig. 1.

**Clinical validation of a high-fidelity prototype**. In two concurrent follow-on pilot studies, the performance of such a 4 channel sensor was evaluated on subjects recruited at the Zuckerberg San Francisco General (ZSFG) Hospital emergency department and Sengupta Hospital and Research Institute (SHRI), yielding 29 and 23 biologically independent samples respectively. These patients over the age of 18, were recruited agnostic of sex, ethnicity etc., strictly based only on the presentation of chest pain under suspicion of Acute Coronary Syndrome. The protocols were reviewed and approved by the corresponding Institutional Review Boards, and all participants signed a written consent form to participate. The patients were selected to represent a wide range of troponin values. Blood was collected in tubes containing lithium heparin, centrifuged at $1000 \times g$ for 10 min. At ZSFG, the plasma was tested for cTnI using a high sensitivity assay on the Siemens Centaur Analyzer[34] (hs-cTnI, Siemens Healthineers). This assay has a LOD of 2.5 ng/L and a 99th percentile of 47 ng/L. In some cases, results from routine troponin testing were used and when that was not available, permission from the patient was granted to collect a fresh blood sample through an intravenous line or venipuncture. At SHRI, CE certified Snibe Maglumi-1000 high sensitivity cTnI assay was employed with an LOD of 0.01 ng/L and a 99th percentile of 19 ng/L. Within 20 min of blood collection for hs-cTnI testing, the 4 channel sensor wearable was installed on the underside of the patient's wrist and left unattended for 5 min.

Infrared readings along with accelerometer data were obtained on a continuous basis. When completed, the sensor was removed, and a file containing these readings were sent through a WiFi connection directly to RCE for data processing.

**Statistical analyses**. Categorical variables were presented as counts and percentages, and continuous variables as means and standard deviations. Receiver operator characteristics (ROC) plot displays performance of a binary classification method with discrete output. Elevated or non-elevated state of troponin was used as the binary output and optical device values as the input for the binary classification method. An R package[35] was used to obtain these ROC plots. In this library, trapezoids are used to compute the AUC indicating an average AUC of 0.853 for a bootstrapped model with sensitivity of 100% and specificity of 70.59%. Confidence intervals are computed with Delong's method with bootstrap resampling[36]. Pearson's correlation coefficient was used to indicate significant linear relationship among quantitative variables and regression analysis was done. A value <0.05 was considered as significant. All statistical tests were performed using Python, R and JMP.

**Reporting summary**. Further information on research design is available in the Nature Research Reporting Summary linked to this article.

## Results and discussion

Absorption spectroscopy is a molecular characterization technique typically used to study the composition of materials and thereby determine concentrations of the substance of interest in its native state. When IR radiation is incident on a material such that the energy is equivalent to the chemical bond vibrational mode[37] of the material, light is absorbed leading to an active vibrational mode. This results in certain energies or wavelengths of light being absorbed by the material that are unique to the material. Thus, the material can be compositionally characterized by performing a differential measurement of the light before and after it passes through the material. Traditional configurations involve directing the infrared radiation through the sample to be measured and detecting the light on the opposite side using a thermal sensor. This restricts the mode of interrogation to in vitro or ex vivo measurement. To overcome this issue, an ATR[38] configuration is adopted where light is totally internally reflected inside an internal reflection element (IRE) or prism of a higher refractive index than the material to be characterized (Fig. 2). Photons come out of the surface of the crystal penetrating the sample and then are coupled back into the system. This partially penetrated evanescent wave can interact with the material on the surface of the crystal, affording the intensities of the frequencies of light measured after passing through the prism to be highly sensitive to the materials present on the surface of the crystal. The penetration depth of the photons is a function of the wavelength of light and the refractive indices of the IRE crystal and sample. In the optical sensor design discussed in this article, MIR radiation is introduced into an IRE (germanium or zinc sulfide) and collected by a detector. The signals before and after the IRE makes contact with the sample are differentially processed to get the optical characteristics of the sample. Figure 2a shows the various vibrational modes observed when MIR light interacts with the epidermis in the ATR configuration. The solution was developed over four phases.

First research step was an ex vivo exploratory phase where the aim was to determine of the ATR configuration was capable of detecting cTnI in blood. This phase involved the testing of blood samples of patients with various cTnI levels on a research grade

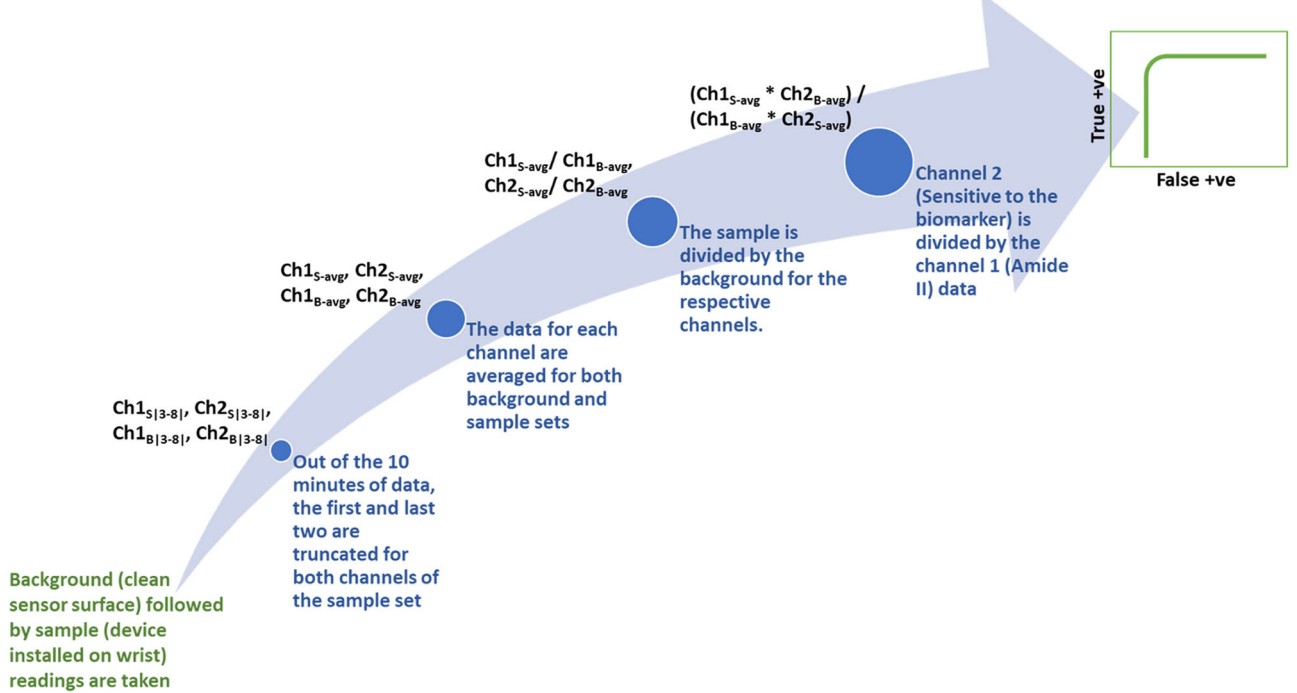

**Fig. 1 Data Analysis Schema.** Schematic of the data analysis pipeline indicating the trajectory from raw optical data collected transdermally to actionable data used for correlation studies. Two channels refer to the intensities obtained from the two optical windows of the detector. Sample set refers to one complete 10 min data stream. (ATR = Attenuated Total Reflectance, SOC = Standard of Care, cTnI = Cardiac Troponin-I).

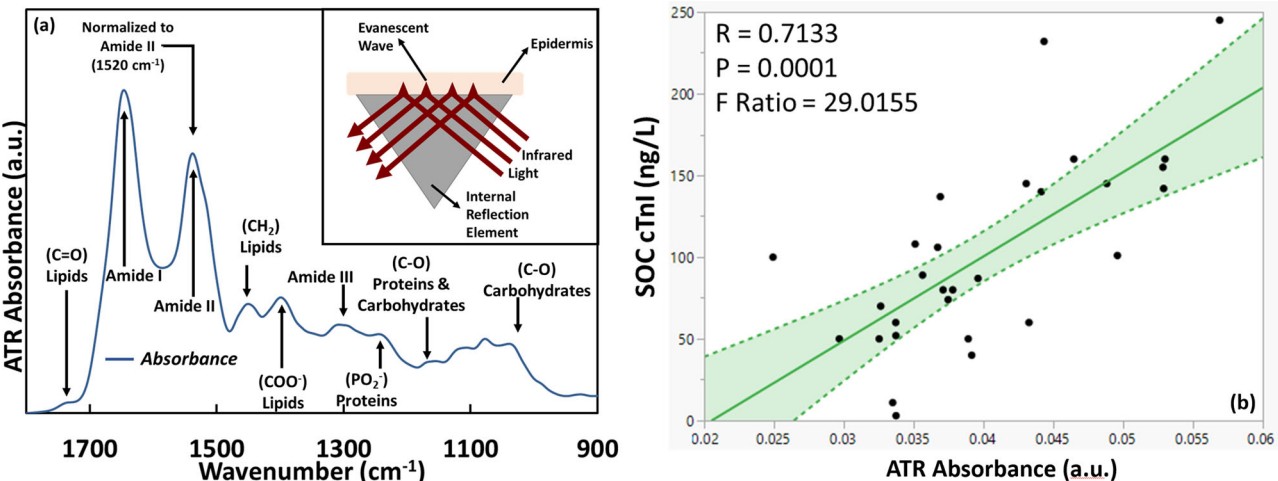

**Fig. 2 Ex vivo exploratory research.** Attenuated Total Reflectance (ATR) absorbance spectrum of epidermis indicating the characteristic peaks representative of the individual molecular components. Inset in (**a**) indicates the working principle of ATR. 1 (**b**) A linear correlation of 71% was observed between the high sensitivity cTnI assay-derived data and the optical device absorbance (n = 30) performed on whole blood (ex vivo). The shaded region indicates the 95% confidence interval. (BNP = Brain Natriuretic Peptide, LAD = Left Anterior Descending, LCX = Left Circumflex, CAG = Coronary Angiogram).

ATR based infrared spectrometer. The optical readouts representing the signature cTnI absorbance peaks were investigated for correlation (Fig. 2b) with that of SOC-derived cTnI concentrations. A positive linear correlation of 71% ($p = 0.0001$) was observed between optical and SOC-derived troponin-I data within the range of 2.5–250 ng/L.

As the next step, the transdermal modality was tested by having patients recruited for coronary angiograms (CAGs) place their thumbs on the surface of a commercial ATR infrared spectrometer. A classification analysis (Fig. 3) based on heterogeneity was performed on 9 different patients. The vertical axis

indicates the level of heterogeneity within the identified cluster. This data processing technique correctly classified four patients as positive for cardiac pathology based on optical biomarker correlations, while also correctly classifying the four normal patients. Coronary Angiogram (CAG) indicated a 50% occlusion in the left anterior descending artery in patient 9 who was clustered with patients 7 and 8 with elevated BNP levels. As revealed by CAG, patient 2 had a 30% occlusion in the left circumflex artery who was also correctly grouped with patients 7, 8, and 9. The heterogeneity of patient 2 data with the rest of the group was inversely proportional to the occlusion in vasculature effectively

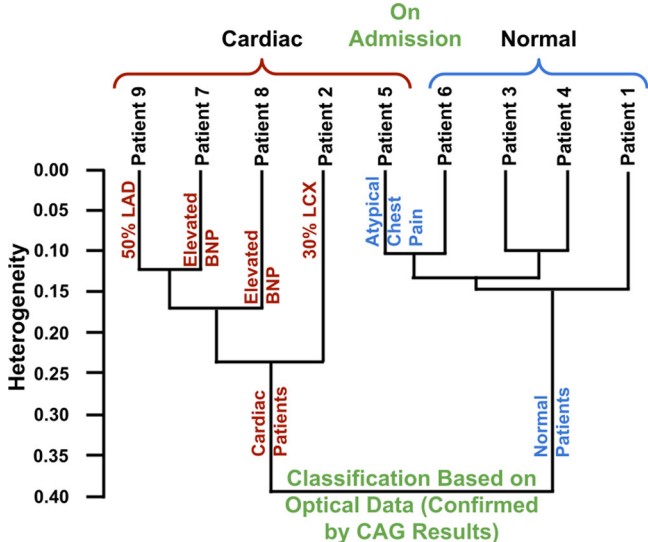

**Fig. 3 Development of a transdermal kit.** Heterogeneity dendrogram based on the hierarchical cluster analysis, indicating the classification of patients based on the noninvasively obtained transdermal optical cardiac protein data in conjunction with coronary angiogram results. (ROC = Receiver Operator Characteristics, AUC = Area Under Curve).

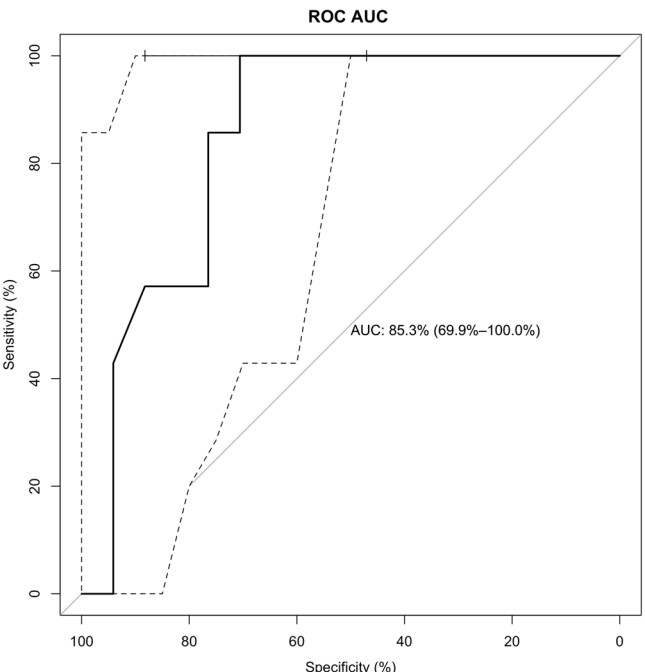

**Fig. 4 Feasibility of point-of-care assessment.** Logistic regression analysis including the non-invasive transdermal optical data and measured troponin-I values yield a sensitivity of 100% and specificity of 70.59%. The receiver operating characteristic curve indicates an average AUC of 0.853 for a bootstrapped model.

demonstrating sensitivity to obstructive coronary artery disease. The optical data-based algorithm stratified patient 5 as normal who was originally admitted as a cardiac patient. This prediction was adjudicated by the cardiologist based on the gold standard CAG who subsequently diagnosed it to be atypical chest pain with no occlusions in the coronary arteries.

As the next step, the feasibility of an ATR spectrometer to be miniaturized into a hand-held point of care device. Such a device was designed and 24 chest pain patients were tested by mounting the device on their wrists. The SOC cTnI concentration served as the ground truth for correlation analysis with the non-invasively obtained optical data. Based on the fourth universal definition of myocardial infarction[39], as informed by the 99th percentile of troponin-I distribution in a reference population, a decision threshold of 19 ng/L is established for elevated troponin-I indicating acute myocardial injury. Upon categorizing the population into two class labels divided by the threshold, a logistical regression analysis (Fig. 4) was conducted between the optical and the binary (elevated and non-elevated) troponin-I data. To prevent overfitting, bootstrap[40] sampling was performed and an AUC of 0.853 (averaged) was observed with a sensitivity of 100% and specificity of 70.59%.

The reason for the lower specificity is thought to be influenced by at least one known factor. As witnessed by the principal investigator, three patients presented with tremors or restlessness which significantly deteriorated the data quality. To prevent this in future studies, the device has been outfitted with a tension-adjustable band to ensure consistent contact between the wrist and the sensing area. Also, an accelerometer was incorporated which will inform the data processing algorithm allowing recovery of some noisy data. Further, there is a possibility that the optical absorbance range primarily used to detect cTnI could have contributions from other biomaterials. The effect of these optical confounders will be reduced in future studies by employing multi-channel optical filters with a narrower range of transmission to the detector thereby improving specificity.

Based on the learnings from the previous phases, a high fidelity POC device was tested for validation in a clinical setting. Figure 5 shows the correlation of this four channel sensor wearable results

against the plasma troponin from both sites with a total sample size of 52. A correlation of $r = 0.7774$ was achieved. A very wide range of hs-cTnI concentrations among the patients were analyzed to determine if sensor exhibited some limitation in the reportable range. While test accuracy for high cTnI values are not required in the diagnosis and management of AMI, it was important to demonstrate that the sensor does not suffer from the "hook effect.", i.e., readings declining with increasing concentrations above a threshold. Typically, the clinical laboratory dilutes samples with an appropriate diluent for analyte concentrations exceeding the upper reportable range of the assay. No inaccurate results were observed for a cTnI concentration up to at least 20,000 ng/L without the need for any dilutions.

In contrast to in vitro, in vivo, ex vivo measurements and implantable devices, the term "on vivo" testing can refer to the use of non-invasive wearable devices[27]. This is not a new concept, as sensors have been used to measure oxygen saturation (pulse oximeter) and total bilirubin (bilirubinometer). Both of these techniques measure pigmented analytes using absorbance within the visible spectrum of light. The troponin sensor uses infrared absorbance and has been shown to have more analytical sensitivity.

After myocardial injury, troponin is released in a variety of forms including ternary complexes (troponin T-I-C), binary complexes (I-C), free submits (T and C), and fragments thereof[41]. At this point in the development of the sensor, it is unclear which forms of circulating troponin are being detected. It is likely that the infrared absorbance signatures will somewhat differ between the various forms affecting the absolute accuracy of the sensor. For this reason, the best medical application for this sensor may be in the early rule out of AMI. Upon clinical consensus, AMI rule-out algorithms have been developed using hs-cTnT[42] and hs-cTnI assays[4]. Accordingly, a diagnosis of AMI can be excluded with either a baseline measurement or after collection of a second sample at 1 h, if a patient presents to the ED with a 3 h history of

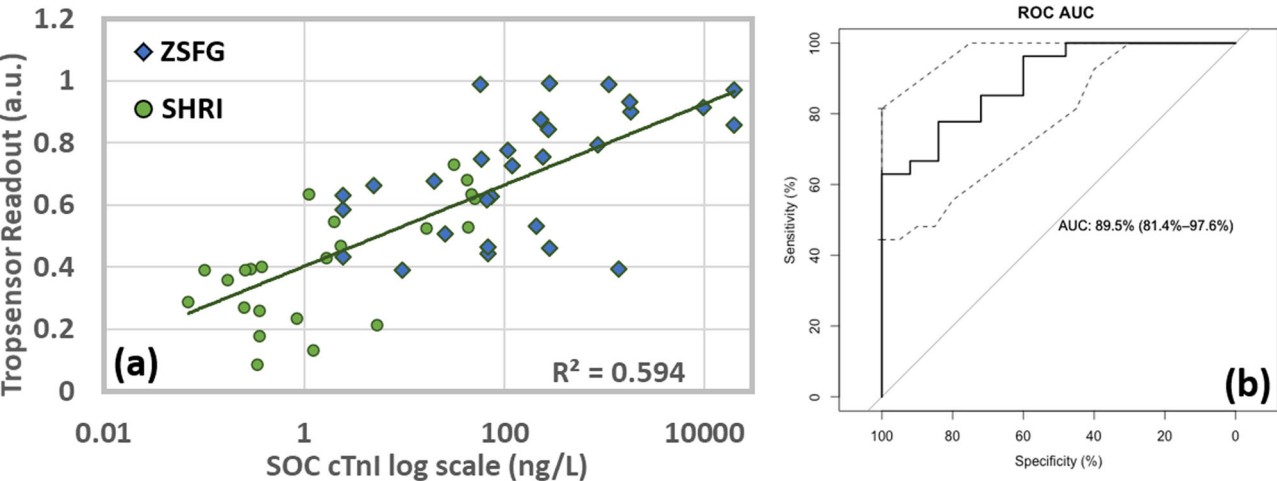

**Fig. 5 Clinical validation of a high-fidelity prototype. a** Correlation of the RCE sensor to plasma troponin using hs-cTnI assays ($n = 52$) at 77%. SOC = standard of care. **b** ROC AUC plot displaying the performance of optical device output toward binary classification of elevated vs non-elevated SOC cTnI levels. AUC—89.5%, sensitivity—62.96% and specificity—100%. (ZSFH = Zuckerberg San Francisco General, SHRI = Sengupta Hospital and Research Institute).

chest pain or longer. The current turnaround time for reporting hs-cTn results from the central laboratory is about 1 h. Point-of-care (POC) testing can reduce the turnaround time to under 30 min, from the time of collection to reporting of results. While there is great interest in developing POC tests for hs-cTn, currently there are no hand-held hs-cTn devices approved for clinical use by the FDA. The use of a sensitive on vivo sensor can produce results within 5 min, without the need of collecting blood and transferring the sample to the testing device. It may be possible to provide continuous readings of cTnI, e.g., at 5 min intervals, if the device is left on the patient. This would enable a caregiver to determine if troponin values are increasing in time. The RCE sensor is portable and self-contained, and therefore testing can be conducted before the patient arrives at the hospital. It remains to be determined if the sensor is sufficiently robust enough to enable accurate testing in other clinical scenarios such as, within an emergency vehicle. When the ATR scheme is employed to interrogate the presence of cardiac biomarkers using infrared light, it is not entirely well understood up to what depth the evanescent light penetrates. However, as made evident by the correlation results, it is clear that the returning evanescent light from the skin back into the IRE surface contains information that is indicative of the presence or absence of cardiac biomarkers. Thus the nature of the measurand or the component of the skin that contains the cardiac biomarkers being interrogated by the light is still unclear. Interstitial space is one of the potential possibilities discussed here. Interstitial fluid is a thin layer of fluid that bathes the cells. This fluid serves as a vehicle for nutrients for the cells. More importantly, they also serve as a conduit for extracellular signaling[43]. It has already been shown that glucose levels can be monitored by sampling the interstitial fluid. Mid-infrared (MIR) spectroscopy is exquisitely sensitive to the chemical composition of the biological materials in the so-called fingerprint region (2000–800 cm$^{-1}$). The penetration depth via the epidermis is too shallow to interrogate the blood capillaries. However, the penetration depth is sufficient to interact with the interstitial fluid. For human skin, the approximate penetration depth is about 1–10 micrometer indicating that the light could potentially interact with the epidermis and the interstitial fluid with the optical sensor surface placed under the wrist (palmar surface). Another possibility for the local source of cardiac proteins is the superficial glands present in the epidermis. Sweat

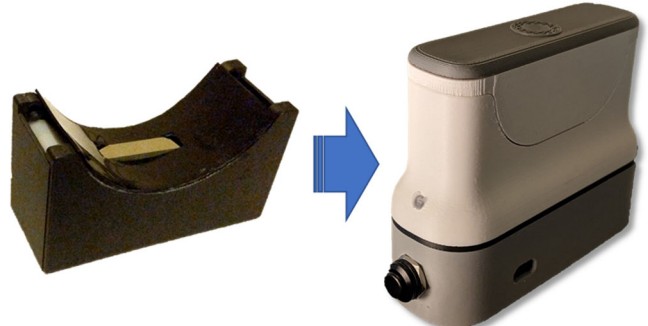

**Fig. 6 Evolution of the non-invasive optical device from the exploratory prototype phase to the current production-ready phase.** The exploratory and prototype phase devices collected two optical data points whereas the current product collects four optical data points, thus potentially improving the specificity of detection.

glands comprise eccrine, apocrine, and apoeccrine glands[44]. It is well understood that a classical symptom of myocardial infarction is profuse sweating or diaphoresis[45]. Sweat comprises many proteins, cytokines, and amino acids. The directed infrared light could interrogate the sweat glands through the epidermis. Sebaceous glands could also contain cardiac proteins diffused via capillaries as they are present directly under the stratum corneum near the hair follicles. Further investigational efforts are needed to establish a source of origin for these biomarkers.

Owing to the modality of optical data collection, the device is hardly susceptible to light pollution and ambient conditions. This is largely mitigated by pulsed measurement and dark background measurement. Considering the fact that the patients recruited irrespective of co-presenting symptoms did not affect the correlation, indicates the potential for the high specificity of the optical technique. The evolution of the transdermal[46] optical device is demonstrated in Fig. 6. Research is currently underway with a device collecting data from tunable optical multi-band to obtain a differential measurement. This will allow a higher correlation to troponin by minimizing the contribution to optical absorbance from other optical confounders in the measurand. The sample preparation for the studies mentioned in this article entailed simply wiping the patient's wrist underside with an alcohol wipe.

To mitigate potential sample preparation confounders, more in-depth studies are being conducted to understand the dependencies of the data quality on tattoos, scars, sweating, etc. A larger sample size study is currently underway to study the effects of various hemodynamic states and clinical confounders such as pulmonary embolism (PE), sepsis and chronic kidney diseases. Although variation in skin pigmentation across subjects has not appeared to have been a confounder in the 76 patients studied transdermally, further in-depth studies will be conducted and reported. Noise handling algorithms and outlier detection techniques will be further evaluated by leveraging deep learning and neural networks.

In conclusion, this letter posits a patient-centric modality for troponin-I monitoring that can inform efficient triaging and timely intervention in the current cardiac clinical workflow. This remote-monitoring-capable technology can be envisioned to empower clinicians in determining the timely clinical course of action to prevent any unnecessary myocardial injury. Furthermore, the use of such non-invasive devices can be extended upstream to facilities such as urgent cares and EMS where blood draws don't often occur.

## Data availability

The datasets generated during and/or analyzed during the current study, including source data for the figures, are available in the Figshare[47] platform at https://doi.org/10.6084/m9.figshare.c.5871056

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

## Author contributions

J.T. and A.B. designed the experiments. J.T. wrote the paper. S.B. performed the data analysis. A.H.B.W. oversaw the 30 patient study at UCSF. S.S. oversaw the 24 patient study at Sengupta Hospital and Research Institute. P.P.S. directed the study and edited the paper.

## Competing interests

P.S. is an advisor to Kencor Health, Ultromics and RCE Technologies and holds option equity with these companies. A.W. is an advisor and holds option equity with RCE Technologies. S.B. holds option equity with RCE Technologies. J.T. and A.B. are employees of RCE Technologies.
