## [Peer Review File · Communications Medicine]

Reviewers' comments:

Reviewer #1 (Remarks to the Author):

Development and Preliminary Validation of an Infrared Spectroscopic Device for Transdermal Assessment of Elevated Cardiac Troponin-I

I have very limited knowledge of spectroscopy and the techniques used in this manuscript. I'm reviewing this paper from a clinical point of view. We take the diagnostic and analytic performance of our troponin assays very seriously. As I understand the authors want to measure cardiac markers such as troponin I transdermally in a clinical setting. I fail to see any evidence that they can do that. Maybe I have missed some part of the manuscript or supplementary files.

Minor points:

Fig 1b: If the measurement data in Fig 1b is the only real validation of their method this must be viewed as preliminary at best. For instance I see two sample with a measured concentration of troponin I of <10 ng/L ng/L that gives an optical absorption of 0.034 whereas one sample with 100 ng/L results in an optical absorption of 0.025. If one only looks at the measurements of samples below 100 ng/L I fail to see any correlation.

The data in Fig 2 is confusing. Where are the measurements of troponin I in this figure?

Reviewer #2 (Remarks to the Author):

The topic in itself is highly interesting and the diagnosis of CHD/myocardial infarction using IR spectroscopy has been proposed. I am amazed that the paper by Petrich et al (Analyst, 134(6):1092-1098, 2009) is not referred to.

- 1) The Abstract is too superficial - it is almost like an overview paragraph. There are no actual results - there should be at least one sentence with hard results. Considering that these patients will have blood being drawn for other reasons - blood drawing is not really that invasive. There is no need for the POC abbreviation in the Abstract - it should be abbreviated first in main manuscript.
- 2) The numbers of patients recruited to this study appears quite small - can the authors specify exactly the study design. If it is the case of the other of 5/so, then this is far too small a cohort to draw any conclusions from.
- 3) Was the diamond facility of the ATR device single bounce or multi-bounce?
- 4) Was the multivariate cluster analysis HCA; please specify.
- 5) My understanding is that the method is reflection, not reflectance?
- 6) What was the signal-to-noise ratio of the spectra like - representative raw and pre-process spectra should be shown.

Reviewer #3 (Remarks to the Author):

In this report, the authors describe the development and preliminary validation of an infrared

spectroscopic device for transdermal assessment of elevated cardiac troponin. Their background nicely highlights the challenges and limitations of current point of care troponin assays and the potential efficiencies that could be gained from a device that could accurately, rapidly and with high sensitivity (comparable to blood-based assays) measure troponin levels. The experiments are proof of concept initial steps and seem to suggest promise--in 9 patients in the initial proof of concept, compared with natriuretic peptide levels and coronary angiography, the IR misclassified one individual who had a 30% lesion at angiography as normal. Given that hsTn is not really used to identify coronary lesions, this is not surprising. In a larger study of 24 patients with a transdermal device more relevant for clinical use, compared with hsTnI at 19 ng/L 99th percentile threshold, the sensitivity and specificity were 100% and 71%, respectively with an AUC of 0.825. A number of technical issues were posited to explain the results providing a foundation for further refinement and development. The conclusions are reasonable and there is a nice summary of "next steps" for development.

The experiments and results are exciting and suggest promise for new POC methods that could improve the efficiency of hsTn testing in the ED and beyond. Suggestions for the authors:

1. The introduction and background are long, choppy and challenging to read with redundancy and not one clear voice/writing style through the various sections. There are a number of typos. It is also not clear why specific manufacturers are referenced in some places rather than more general statements. If there is significance to the specific manufacturers that are called out, this should be explained in detail.
2. For the clinical studies, particularly the 24 patient comparison of the transdermal wearable with laboratory based hsTnI, it would be helpful to have a table of baseline characteristics of the participants. In particular, one question for a device that measures a molecule by spectroscopy through the skin is whether the degree of skin pigmentation affects transmission or results. Testing in African Americans (or other groups) would be important and/or an explanation as to why this methodology would not be expected to be affected would be useful.

Reviewers' comments:

Reviewer #1 (Remarks to the Author):

Development and Preliminary Validation of an Infrared Spectroscopic Device for Transdermal Assessment of Elevated Cardiac Troponin-I

I have very limited knowledge of spectroscopy and the techniques used in this manuscript. I'm reviewing this paper from a clinical point of view. We take the diagnostic and analytic performance of our troponin assays very seriously. As I understand the authors want to measure cardiac markers such as troponin I transdermally in a clinical setting. I fail to see any evidence that they can do that. Maybe I have missed some part of the manuscript or supplementary files.

We thank the reviewer for the opportunity to clarify our work. We agree that the work provides early steps towards the development of a technology for measuring Troponin I transdermally in clinical settings. We used a translational approach in which we first determined the optical signature of troponin-I for potential clinical application (**page 5, line 109**). Second, the blood samples from patients with elevated troponin-I were characterized (Figure 1b) using the proposed optical technique while also assayed using the conventional reagent-based Chemiluminescent Immunoassay which is standard-of-care (**page 6, line 117**). Thirdly, the optical technique was used **transdermally** on patients presenting with chest pain for whom the troponin-I was elevated. The **non-invasively obtained optical data** was compared with the conventional reagent-based Chemiluminescent Immunoassay which is standard-of-care (Figure 3). We observed a modest correlation between the two modalities, and an **AUC of 0.8526** for predicting a binary outcome of elevated Troponin I (**page 7, line 151**). We agree that future work in larger cohorts may help translating the technology into clinical practice.

Minor points:

Fig 1b: If the measurement data in Fig 1b is the only real validation of their method this must be viewed as preliminary at best.

We agree that these observations are preliminary in nature as stated above. While Fig 1b correlation data is modest, figure 3 shows better consistency for range of troponin I values which may be clinically useful.

For instance, I see two sample with a measured concentration of troponin I of <10 ng/L ng/L that gives an optical absorption of 0.034 whereas one sample with 100 ng/L results in an optical absorption of 0.025. If one only looks at the measurements of samples below 100 ng/L I fail to see any correlation.

The reviewer is correct in his observation, as expected the correlations of the absolute values are modest; however, the consistency of the information improves while considering intervals as highlighted below and in Figure 3.

The data in Fig 2 is confusing. Where are the measurements of troponin I in this figure? Data in figure 2 represents the hierarchical cluster analysis (HCA) when considering the coronary angiogram (CAG) based patient outcome and the non-invasive optical data in the wavelength range covering the unique absorption features relating to h-FABP, Cardiac Troponin-I, and BNP (lines). The figure caption has been modified to provide better clarity (**page 7, figure 2**).

Reviewer #2 (Remarks to the Author):

The topic in itself is highly interesting and the diagnosis of CHD/myocardial infarction using IR spectroscopy has been proposed. I am amazed that the paper by Petrich et al (Analyst, 134(6):1092-1098, 2009) is not referred to.

Thank you for suggesting us to include this excellent reference. This reference has been included (**page 3, line 68**).

1) The Abstract is too superficial - it is almost like an overview paragraph. There are no actual results - there should be at least one sentence with hard results.

The abstract has been revised as suggested (**page 2, lines 20-34**).

Considering that these patients will have blood being drawn for other reasons - blood drawing is not really that invasive.

One of the biggest advantages over the SOC serial draws is the availability of continuous troponin data that provides the physician an additional context of trending cTn levels. Although it is true that blood will be drawn for other reasons, this optical non-invasive troponin-I measurement tool can be incredibly important in quicker triaging of patients outside the confines of emergency rooms where blood sampling and measurements are not possible. Furthermore, the use of such non-invasive devices can be extended upstream to facilities such as urgent cares and EMS where blood draws don't often occur (**page 12, line 231**).

There is no need for the POC abbreviation in the Abstract - it should be abbreviated first in main manuscript.

The abbreviation has been removed from the abstract and introduced in the main body as pointed out (**page 3, line 47**).

2) The numbers of patients recruited to this study appears quite small - can the authors specify exactly the study design. If it is the case of the other of 5/so, then this is far too small a cohort to draw any conclusions from.

We agree this is a 'proof-of-concept' study where data from total 62 human subjects were assessed. This included **blood-based** sampling from a cohort of 30 patients recruited from the emergency department at Zuckerberg San Francisco General Hospital. These patients were under suspicion of acute coronary syndrome (ACS), and standard of care blood based Troponin was ordered.

The second analysis (**transdermal, non-invasive**) included 8 patients, four of these presented with chest pain and underwent coronary angiograms. Remaining patients were recruited when they presented at the hospital for non-cardiac conditions.

The third study (**transdermal, non-invasive**) comprised 24 patients who were recruited from the CCU where they presented as ACS.

3) Was the diamond facility of the ATR device single bounce or multi-bounce?

In the study, the diamond ATR prism allows a single bounce of the IR light (**page 6, line 112**).

4) Was the multivariate cluster analysis HCA; please specify.

The multivariate analysis performed was a Hierarchical Cluster Analysis (HCA). This has been indicated in the figure caption and body (**page 6, line 133 and page 7, figure 2 caption**).

5) My understanding is that the method is reflection, not reflectance?

The ATR scheme works on the principle of total internal **reflectance**. Since ATR produces an evanescent wave, where the photons are either **absorbed or transmitted** by the material (skin) in contact with the ATR prism surface thereby resulting in an absorbance spectrum (and not absorption, as absorption requires the depth of interaction). So we are not really looking at the light that is reflected but rather absorbed by the skin.

6) What was the signal-to-noise ratio of the spectra like - representative raw and pre-process spectra should be shown.

Due to the intrinsic property (*Fellgett's advantage) of ATR-FTIR, the signal-to-noise ratio is in the order of 10,000:1. For example, the following chart shows the **raw, unprocessed spectral data** of human whole blood repeated 3 times (red, blue and orange). Each has 32 co-added scans. The standard deviation is so low that it is smaller than the plot's linewidth after normalization and averaging.

Furthermore, the effects of SNR in all the studies reported are encompassed in the correlation/receiver operator analysis.

(*Fourier transform infrared spectroscopy. Part II. Advantages of FT-IR, J. Chem. Educ. 1987, 64, 11, A269)

Reviewer #3 (Remarks to the Author):

In this report, the authors describe the development and preliminary validation of an infrared spectroscopic device for transdermal assessment of elevated cardiac troponin. Their background nicely highlights the challenges and limitations of current point of care troponin assays and the potential efficiencies that could be gained from a device that could accurately, rapidly and with high sensitivity (comparable to blood-based assays) measure troponin levels. The experiments are proof of concept initial steps and seem to suggest promise--in 9 patients in the initial proof of concept, compared with natriuretic peptide levels and coronary angiography, the IR misclassified one individual who had a 30% lesion at angiography as normal. Given that hsTn is not really used to identify coronary lesions, this is not surprising. In a larger study of 24 patients with a transdermal device more relevant for clinical use, compared with hsTnI at 19 ng/L 99th percentile threshold, the sensitivity and specificity were 100% and 71%, respectively with an AUC of 0.825. A number of technical issues were posited to explain the results providing a foundation for further refinement and development. The conclusions are reasonable and there is a nice summary of "next steps" for development.

We thank the reviewer for the encouraging comments.

The experiments and results are exciting and suggest promise for new POC methods that could improve the efficiency of hsTn testing in the ED and beyond. Suggestions for the authors:

1. The introduction and background are long, choppy and challenging to read with redundancy and not one clear voice/writing style through the various sections. There are a number of typos. The introduction and background are re-written for better conciseness and readability. It is also edited for typographical errors and consistency of voice style.

It is also not clear why specific manufacturers are referenced in some places rather than more general statements. If there is significance to the specific manufacturers that are called out, this should be explained in detail.

Since there are differences in the device parameters such as coefficient of variance, sensitivity, diagnostic threshold, etc., the specific manufacturers are called out for the sake of reproducibility of the research. However, some manufacturers mentioned previously have been removed for the manuscript and replaced with a reference (page 2, reference 4).

2. For the clinical studies, particularly the 24 patient comparison of the transdermal wearable with laboratory based hsTnI, it would be helpful to have a table of baseline characteristics of the participants.

The correlation analysis reported here doesn't consider the eventual patient outcome. Rather, the absolute standard-of-care assay measured troponin-I values and non-invasive transdermal optical values are compared and analyzed. The participant characteristics are now included in the methods addendum (also shown below).

Patient number	Patient Demographics Age (years)	Diabetes	Hypertension	On analysis (CKD/ESRD)	Smoking/Tobacco Chewing	Hypercholesterolemia	Symptoms at admission	Presenting Clinical Sx/Differentials	Prior History of Cardiac Disease	Previous Stress Test (if available)	Current CAG Findings	Current CAG (intervention)	Current Echo Ejection Fraction	Current Echo (Wall Motion Abnormalities)	Current Stress Echo (Color Kinetics)	Lab measurements (mg/dL)	Tropensor Optical readout (µg/L)	Diagnosis	Medications		
																				0 = None 1 = CAD 2 = Heart Failure 3 = Arrhythmia 4 = Cardiomyopathies 5 = Structural Heart Disease 6 = Other Sx (Specify)	0 = None 1 = Thrombolytics 2 = PCI 3 = CABG
1	62	M	0	0	0	1	0	Abdominal Trauma patens	4	0	NA	Not done	-	50%	1	1	0.01	0.839680135	4	4.7	
2	68	M	0	1	0	1	0	Chest Pain with Breathlessness Diff Dx: Unstable Angina CHF	3	1	NA	Triple Vessel disease with 1st Main	Adv - CABG	45%	1	1	3.1	17000	0.913169089	2	1,2,5,7
3	59	M	1	1	0	0	0	Diff Dx: HT/DM/HD/Post PTCA/ Unstable Angina, admitted with chest pain	3	1	1	stent in RCA with TIM II Flow POA & RV	Not required	50%	1	1	0.01	0.839181855	4	1,2,3,5,7	
4	65	F	0	1	0	0	0	Diff Dx: HT / HPEFF / Hypothyroid	3	0	NA	Normal coronaries	Not required	50%	1	1	0.01	745.8	0.85204345	8	1,3,5,7
5	69	F	1	1	0	0	0	Diff Dx: HT / DM - Normal	3	0	NA	Not done	-	55%	0	0	0.01	0.857736297	3	4,5,7	
6	86	F	1	1	0	0	0	Diff Dx: DCM/DM /LVF 35%	4	0	NA	Not done	-	35%	1	0	0.01	0.932051631	4	1,3,5,7	
7	67	M	0	1	0	0	0	Diff Dx: HT / LVF 30%/ Acute Coronary Syndrome	4	1	NA	post PTCA to LAD .LAD 90% , RCA 90%	Not required	30%	1	0	21.63	1111.7	0.980432131	4	1,3,5,7
8	60	F	1	1	0	0	0	Diff Dx: HT/DM/MI/valvulopathy A/MI/EF 40%	1	0	NA	LAD 90%	PTCA to LAD done	40%	1	1	21.5	2200	0.921454412	1	1,3,5,7
9	75	M	0	0	0	1	0	Anterior Wall MI	1	0	NA	CAG not done as patient was discharged against medical advice	-	40%	1	1	1.33	1129.2	0.8513383	1	1,3,5,7
10	77	F	0	1	0	0	0	HT/HD/ Acute Diabetic Heart Failure, admitted with chest pain, severe breathlessness, swelling at chest body	6- Acute heart failure	0	not available	discharged without doing CAG	-	60%	0	0	0.42	918.7	0.972376887	5	1,3,5,7
11	51	M	1	1	0	0	0	Admitted with congestive chest pain, known case of HT / DM / Old CVA / Seizure disorder	Seizure disorder with acute LVF	0	not available	Mid LAD 20% flow	not required	50%	1	1	0.09	799.7	0.828598208	5	1,3,5,7
12	72	M	0	1	0	0	0	Chest pain , chest heaviness , breathlessness , nausea , vomiting	IVM	0	not available	LAD - 80% , RCA 100%	PLAN for PTCA to RCA	45%	1	1	22.2	2407	0.969548355	1	1,7,9
13	85	M	0	1	0	1	0	chest pain , breathlessness , uneasiness	3	0	not available	LAD 90% , RCA 90% , OMT 90%	-	50%	1	1	0.45	1200.1	0.82840254	3	
14	43	M	0	1	0	0	0	Chest pain , breathlessness, uneasiness, sweating	1	0	not available	LAD 90%	PTCA to LAD	50%	1	1	50	679.4	0.896194966	1	1,2,3,5,7
15	73	M	1	1	0	0	0	profuse sweating, uneasiness, breathlessness	3	1	1	LAD 80% L/CX 80% RCA - 50%	PLAN FOR CABG	50%	1	1	0.06	466.9	0.76267331	5	1,4,7,9
16	80	F	1	1	0	0	0	chest pain , anxiety , breathlessness , palpitation	3	0	1	LAD - 90% , OMT 90%	PTCA TO LAD	45%	1	1	0.13	20816.3	0.816781046	3,5	1,3,4,5,7,10
17	59	M	1	1	0	1	0	chest pain , breathlessness , sweating, uneasiness	1	1	1	2,3,4-c	CABG	50%	1	1	50	886.9	0.988888408	1	1,3,7,10,2
18	63	F	0	1	0	0	0	chest pain , uneasiness, restlessness	1	1	1	patient stent in LAD OI thrombus	medical management	50%	1	1	21.3	2180	0.92434048	1	1,3,7,5,10
19	85	M	0	1	0	0	0	uneasiness , breathlessness	2,3	1	1	patient not doing CAG/ LCA 80% , OMT 90% , RCA 100%	PTCA TO L/CX , OM RCA	45%	1	1	0.01	3683.9	0.766360271	3	1,4,5,7,9
20	73	M	1	1	0	0	0	chest pain , uneasiness, breathlessness	3	1	1	1,2,3,4-b	CABG advised	35%	1	0	1.34	7981.8	0.876747634	3	1,3,5,7,9
21	74	F	1	1	0	0	0	Chest pain , uneasiness, anxiety sweating	3	1	1	1,2,3,4-b	CABG advised	35%	1	0	1.34	7981.8	0.876747634	3	1,3,5,7,9
22	70	F	1	1	0	0	0	uneasiness, breathlessness, sweating	3	1	1	2-c,3-b	not done	40%	1	0	1.54	1193	1.000909183	3	1,2,3,9,7
23	32	M	0	0	0	1	0	chest pain , breathlessness	1	0	0	2-c	PTCA to LAD	35%	1	0	50	1284.3	0.987978206	1	1,5,9,7
24	75	M	0	1	0	0	0	chest pain , uneasiness , sweating	3	0	0	not done as creatinine value is 2.2	-	50%	1	1	0.38	12745.15	0.885250247	3	1,6,9,7

In particular, one question for a device that measures a molecule by spectroscopy through the skin is whether the degree of skin pigmentation affects transmission or results. Testing in African Americans (or other groups) would be important and/or an explanation as to why this methodology would not be expected to be affected would be useful.

We agree that values in patients with different skin colors is important. Right now the data is derived from Caucasians and Indian population (who have darker skin pigmentation). We agree that we need to conduct more studies in the future and report on the potential confounders including skin pigmentation. This has been acknowledged (page 12, line 223).

Reviewers' comments:

Reviewer #1 (Remarks to the Author):

No comments.

Reviewer #2 (Remarks to the Author):

This is nice work and very promising.

Reviewer #3 (Remarks to the Author):

The authors have adequately addressed my initial review and comments. I have no further comments or suggestions for the authors.

Reviewer #4 (clinical chemistry, cardiovascular biomarkers, troponin assays) (Remarks to the Author):

The purpose of this study is to assess the capability of transdermal FTIR spectroscopy to identify elevated circulating cardiac troponin concentrations.

Major comments:

- Since this is a proof-of-concept study, I would expect the authors to present more raw data, including FTIR spectra of the cardiac biomarkers studied (cardiac troponin, BNP, FABP3) and troponin concentration vs optical absorbance plots (analogous to Fig 1) for wearable device data currently summarized in Figure 3.
- The data presented have been heavily analyzed (e.g., hierarchical cluster analysis in Figure 2) and the work could not be reproduced by others without significantly more experimental and data analysis method being presented.
- As an example of the lacking experimental details, the authors state on page 7: "The optical readouts were investigated for correlation (figure 1b) with that of SOC-derived cTnI concentrations." The "optical readouts" referred to are not described in any further detail but form the basis of the entire study. As presented currently, the reader must implicitly trust that the authors are measuring cardiac troponin transdermally and are not simply correlating non-specific absorption spectra with circulating cardiac troponin concentrations. I am not implying the ATR FTIR method employed lacks specificity for cardiac troponin, but that the paper has not provided any direct evidence demonstrating specificity—only correlational data, having gone through extensive processing (not presented), are available to the reader.
- Multivariate cluster analysis presented in Fig 2 makes use of 3 cardiac biomarker measurements: the authors do not clarify how they were able to distinguish these three biomarkers. Given that the paper is otherwise focused on cardiac troponin, I wonder how well the cluster analysis would classify cardiac vs normal patients using cardiac troponin values only.

Minor comments

- The rebuttal letter's response to Reviewer 1 presents optical absorbance vs cardiac troponin

concentration data not included in the manuscript—I think these plots are very informative, as they demonstrate the concentration-dependence of the correlation observed. While correlations at low troponin concentrations are poor, the fact that correlation improves dramatically at higher concentrations is encouraging news for this proof of concept study.

- The rebuttal letter draws a distinction between absorbance and absorption spectra, indicating that the former is the appropriate characterization of ATR FTIR data collected. However, Figure 1 uses optical absorption on the x-axis label and the term “absorption” is used throughout the manuscript.

- Figure 4 cannot be readily understood without more experimental details provided elsewhere in the manuscript. Terms used without context or definition in the figure include: channels, sample set, sensitive to the biomarker

- Page 6 refers to troponin I values “between the limits of detection of the SOC assay...”. This statement does not make sense as written. The limit of detection refers to the lowest concentration that can be distinguished from 0. There is only 1 limit of detection. I wonder if the authors intend to say that values were above the limit of detection and within the analytical measurement range of the standard of care troponin assay.

- The authors utilize both a globally employed high-sensitivity cardiac troponin I assay (Siemens Advia Centaur) and a novel high sensitivity troponin assay not available in the US market and I suspect not available globally in general (Snibe Maglumi-1000). The authors should acknowledge the status of the latter instrument (e.g., research use only, approved by some national regulatory body, etc.) as I suspect most readers will be unfamiliar with this instrument.

- Page 9 refers to a decision threshold of 19 ng/L to establish myocardial infarction diagnosis. This statement should be updated to use the term “myocardial injury” as opposed to infarction, given that elevated cardiac troponin is not specific for an infarct but rather myocardial injury (necrosis) from any source.

- On page 6, the authors use the term “high sens-cTnI”. This should be updated to “high sensitivity”.

Reviewers' comments:

Reviewer #1 (Remarks to the Author):

No comments.

Reviewer #2 (Remarks to the Author):

This is nice work and very promising.

Reviewer #3 (Remarks to the Author):

The authors have adequately addressed my initial review and comments. I have no further comments or suggestions for the authors.

Reviewer #4 (clinical chemistry, cardiovascular biomarkers, troponin assays) (Remarks to the Author):

The purpose of this study is to assess the capability of transdermal FTIR spectroscopy to identify elevated circulating cardiac troponin concentrations.

Major comments:

- Since this is a proof-of-concept study, I would expect the authors to present more raw data, including FTIR spectra of the cardiac biomarkers studied (cardiac troponin, BNP, FABP3) and troponin concentration vs optical absorbance plots (analogous to Fig 1) for wearable device data currently summarized in Figure 3.

We thank the reviewer for the opportunity to clarify our work. In this revision, we have included the results from another independent study at SHRI to obtain an aggregated sample of 52. While designed as a proof of concept, we hope that this will address any concerns regarding the feasibility of this new transdermal point-of-care troponin assay. **Page 11, line 207.**

The spectral data shown in Fig 1 is obtained from a Fourier Transform Infrared (FTIR) Spectrometer which scans through the entire midinfrared wavelength range. This allowed us to identify specific wavelengths of interest for the detection of troponin. The optical POC device was designed to operate only on two wavelength windows that were identified in the previous step to be relevant for troponin detection. Further, the intensities from these two windows are ratioed for internal standardization. So in essence, the optical device produces only one data point per measurement, as opposed to multiple datapoints (spectral sweep) obtained from an FTIR spectrometer. Subsequently, each datapoint (representative of a measurement) is plotted against the standard of care obtained troponin value. From this scatter plot, a correlation curve is established which is an indicator of the accuracy with which the actual troponin concentration in the system can be predicted if the optical device datapoint is known. This is shown in Fig 5a in the current revised manuscript.

With regards to the spectral data indicated in figure 1, the wearable version of the optical device produces only 2 absorbance values relevant to troponin, which are ratioed with background as per typical practice. The entire pipeline for processing is detailed in figure 4. **Page 10, figure 4.**

- The data presented have been heavily analyzed (e.g., hierarchical cluster analysis in Figure 2) and the work could not be reproduced by others without significantly more experimental and data analysis method being presented.

We intend to present how the technology evolved and agree that the initial exploratory analysis included is not detailed. In the revised version, therefore the evolution of technology is presented in four steps: 1) Ex vivo exploratory research, 2) Development of a transdermal kit 3) Feasibility of point-of-care assessment and finally, 4) clinical validation of a high-fidelity prototype. The cluster analysis in small sets of patients represents step #2 which was used for development of the transdermal kit. However, subsequent confirmation of the initial observations with correlation analysis presented in 52 patients from two medical centers helps explain the foundation of how the technological assessments were carried out. As detailed in the text, segments of the spectral data where the contribution from troponin absorbance is higher are used to classify the patients. No other preprocessing is done. The details of the algorithm is mentioned in the text. **Page 7, line 145.**

- As an example of the lacking experimental details, the authors state on page 7: “The optical readouts were investigated for correlation (figure 1b) with that of SOC-derived cTnI concentrations.” The “optical readouts” referred to are not described in any further detail but form the basis of the entire study. As presented currently, the reader must implicitly trust that the authors are measuring cardiac troponin transdermally and are not simply correlating non-specific absorption spectra with circulating cardiac troponin concentrations. I am not implying the ATR FTIR method employed lacks specificity for cardiac troponin, but that the paper has not provided any direct evidence demonstrating specificity—only correlational data, having gone through extensive processing (not presented), are available to the reader.

We thank the reviewer for the opportunity to clarify. While a direct measurement of troponin is not ascertained here, we have increased the sample size of our correlation studies in the revised submission with appropriate selection criteria and also developed diagnostic models that indicate the ability to track the diagnostic thresholds of troponin even if it is indirectly measured. However, we do agree that further studies are required to validate the quantification of troponin concentration taking into account other potential clinical and optical confounders and specified this in the discussion.

- Multivariate cluster analysis presented in Fig 2 makes use of 3 cardiac biomarker measurements: the authors do not clarify how they were able to distinguish these three biomarkers. Given that the paper is otherwise focused on cardiac troponin, I wonder how well the cluster analysis would classify cardiac vs normal patients using cardiac troponin values only.

The presence of troponin should largely contribute to the absorbance spectral segment used for the cluster analysis. However, other analytes could also have contributions albeit minor. The text has been modified to reflect the data segment used. **Page 7, line 143.**

Minor comments

- The rebuttal letter’s response to Reviewer 1 presents optical absorbance vs cardiac troponin concentration data not included in the manuscript—I think these plots are very informative, as they demonstrate the concentration-dependence of the correlation observed. While correlations at low troponin concentrations are poor, the fact that correlation improves dramatically at higher concentrations is encouraging news for this proof of concept study.

We agree with the premise of the reviewer's comment. Please note that this refers to the blood based optical measurements which is part of our 1st out of 4 phases of development. These plots were intended as a response to the previous reviewer indicating that the correlation at any smaller subset of the troponin range is always dependent on the correlation fit of the entire range. Hence the fit improves as the range is increased (simultaneously increasing the sample size). Moreover, the plot in Fig 1b also shows the shaded 95% confidence interval for the entire range, essentially conveying the same message. **Page 6, Fig 1b and Page 16, line 301.**

- The rebuttal letter draws a distinction between absorbance and absorption spectra, indicating that the former is the appropriate characterization of ATR FTIR data collected. However, Figure 1 uses optical absorption on the x-axis label and the term "absorption" is used throughout the manuscript.

We thank the reviewer for pointing out this discrepancy. The figure and the text have been modified accordingly. **Page 6, figure 1.**

- Figure 4 cannot be readily understood without more experimental details provided elsewhere in the manuscript. Terms used without context or definition in the figure include: channels, sample set, sensitive to the biomarker

These terms have now been clarified in the figure caption. **Page 10, figure 4.**

- Page 6 refers to troponin I values "between the limits of detection of the SOC assay...". This statement does not make sense as written. The limit of detection refers to the lowest concentration that can be distinguished from 0. There is only 1 limit of detection. I wonder if the authors intend to say that values were above the limit of detection and within the analytical measurement range of the standard of care troponin assay.

The values included in the study were between the "upper and lower" limits of detection of the assay. This clarification has been made in the text. **Page 6, line 130.**

- The authors utilize both a globally employed high-sensitivity cardiac troponin I assay (Siemens Advia Centaur) and a novel high sensitivity troponin assay not available in the US market and I suspect not available globally in general (Snibe Maglumi-1000). The authors should acknowledge the status of the latter instrument (e.g., research use only, approved by some national regulatory body, etc.) as I suspect most readers will be unfamiliar with this instrument.

The Snibe Maglumi-1000 is an assay used outside of the US carrying a CE mark. This has been clarified in the text. **Page 9, line 173.**

- Page 9 refers to a decision threshold of 19 ng/L to establish myocardial infarction diagnosis. This statement should be updated to use the term "myocardial injury" as opposed to infarction, given that elevated cardiac troponin is not specific for an infarct but rather myocardial injury (necrosis) from any source.

The reviewer's comment is appreciated and the text is modified to that effect. **Page 10, line 187.**

- On page 6, the authors use the term "high sens-cTnI". This should be updated to "high sensitivity".

This correction has been made in the manuscript text. **Page 6, line 128.**

REVIEWERS' COMMENTS:

Reviewer #4 (Remarks to the Author):

The authors have addressed many of my concerns. I still have a few suggestions that I think should be addressed:

-Phase I (Ex vivo exploratory research) page 6 indicates that cardiac troponin I, CK-MB, and BNP were optically characterized in their pure form. The material source/supplier and product codes should be specified here.

-Page 3 notes that POC assays suffer from low accuracy, making them incompatible with rapid rule-out algorithms. Poor sensitivity is the primary issue here, not accuracy, so this statement should be updated to indicate that POC assays suffer from low sensitivity.

-Page 6 uses the term limit of detection inappropriately. Limit of detection refers to the lowest concentration that can be reliably distinguished from 0 concentration. The statement in question should be updated to something like "...between the limit of detection and upper reporting limit."

-Page 7 refers to the heterogeneity of patient 2 data. Please include a reference to Figure 2 to assist the reader in interpreting this statement.

-Page 9 indicates that a particular optical range was selected such that cTnI would have the largest contribution to the absorbance. This optical range should be clarified (at minimum, width of the range) and the authors should clarify here that they have not yet demonstrated specificity of this range for cTnI (as opposed to cTnT or other troponin isoforms).

-Page 16: the sentence beginning "However, the penetration depth..." includes a period mid-sentence and requires revision.

Reviewer's comments

Reviewer #4:

Remarks to the Author:

The authors have addressed many of my concerns. I still have a few suggestions that I think should be addressed:

We greatly appreciate the reviewer's investment by critiquing this communication. The presentation of our work has been greatly improved by incorporating the following suggestions. Our responses are indicated in red font.

-Phase I (Ex vivo exploratory research) page 6 indicates that cardiac troponin I, CK-MB, and BNP were optically characterized in their pure form. The material source/supplier and product codes should be specified here.

The material source and the product codes have been specified in the main text. (Page 5, lines 112-114)

-Page 3 notes that POC assays suffer from low accuracy, making them incompatible with rapid rule-out algorithms. Poor sensitivity is the primary issue here, not accuracy, so this statement should be updated to indicate that POC assays suffer from low sensitivity.

We agree that this is an important distinction and the text has been modified accordingly. (Page 4, line 69)

-Page 6 uses the term limit of detection inappropriately. Limit of detection refers to the lowest concentration that can be reliably distinguished from 0 concentration. The statement in question should be updated to something like "...between the limit of detection and upper reporting limit."

The text has been modified as per the suggestion. (Page 6, lines 123 - 124)

-Page 7 refers to the heterogeneity of patient 2 data. Please include a reference to Figure 2 to assist the reader in interpreting this statement.

The reference to the figure has been added. (Page 11, line 245)

-Page 9 indicates that a particular optical range was selected such that cTnI would have the largest contribution to the absorbance. This optical range should be clarified (at minimum, width of the range) and the authors should clarify here that they have not yet demonstrated specificity of this range for cTnI (as opposed to cTnT or other troponin isoforms).

As per the reviewer's suggestion, the width of the range has been included. A sentence to clarify the specificity for cTnI has also been added. (Page 6, lines 137 - 138)

-Page 16: the sentence beginning "However, the penetration depth..." includes a period mid-sentence and requires revision.

The typographical error has been rectified to now state "...penetration depth is sufficient to interact with the interstitial fluid. For human skin, the approximate ..." (Page 14, line 329)